# Loss of Caprine Arthritis Encephalitis Virus (CAEV) Herd Accreditation: Characteristics, Diagnostic Approach, and Specific Follow-Up Scenarios on Large Dairy Goat Farms

**DOI:** 10.3390/pathogens11121541

**Published:** 2022-12-15

**Authors:** Karianne Peterson, René van den Brom, Marian Aalberts, Carlijn ter Bogt-Kappert, Piet Vellema

**Affiliations:** 1Department of Small Ruminant Health, Royal GD, P.O. Box 9, 7400 AA Deventer, The Netherlands; 2Department of Research and Development, Royal GD, P.O. Box 9, 7400 AA Deventer, The Netherlands

**Keywords:** small ruminant lentivirus, SRLV, caprine arthritis encephalitis virus, goat, bulk milk, ELISA

## Abstract

The retrovirus causing caprine arthritis encephalitis (CAE), a slowly progressive inflammatory disease in goats, belongs to the group of small ruminant lentiviruses (SRLVs) which cause lifelong infections that ought to be avoided for animal welfare as well as economic reasons. SRLV accreditation has been in place for forty years in The Netherlands and is based on the screening of small ruminant sera for specific antibodies. This paper evaluates 38 dairy goat herds that lost CAEV accreditation between 2012 and 2022. The characteristics of these herds are discussed, and specific follow-up scenarios, depending on desired goals, are introduced. The herd size of the participating herds varies from approximately 400 to 4600 adult dairy goats. The larger herds tended to be more prone to lose herd accreditation and had more difficulties regaining accreditation. Possible routes of introduction are lined up. The Royal GD’s tailor-made approach and advice to support livestock farmers with herds that have lost CAE accreditation are discussed in detail. Specific emphasis is placed on the strategic deployment of various diagnostic tests (such as antibody ELISAs and PCR) in different media, such as (pooled) sera, (bulk)milk and tissue samples. Special attention is paid to the added value of retrospective bulk milk testing or the specific testing of groups based on housing and management, which enables the investigation of the moment of viral introduction and route of transmission into a herd. Furthermore, the prospective implementation of bulk milk and strategic pooled milk sample testing in the Dutch SRLV accreditation programs intensifies surveillance and enables the taking of swift action to prevent further transmission within and between herds. An appeal is made to share experiences to improve programs collectively, and to start research into the underlying mechanisms.

## 1. Introduction

Over the last decades, the dairy goat industry in The Netherlands has become a serious agricultural sector which has grown significantly. In 2020, on average 670,842 goats were kept on 14,730 registered locations. The average herd size of the 400 professional dairy goat farms has increased to 1411 goats [1]. Some producers have even expanded numbers to more than eleven thousand.

Within the dairy goat industry, most farmers have their herds accredited for caseous lymphadenitis (CLA) [2] and caprine arthritis encephalitis (CAE) [3]. Both diseases can have a huge impact on animal welfare, production, trading and thus economics. The small ruminant lentiviruses caprine arthritis encephalitis virus (CAEV) and maedi visna virus (MVV) show high (anti)genic variability, and have the ability to wreak havoc on goat and sheep populations worldwide. They can cause incurable chronic, progressive, complex neurological, arthritic, pulmonary and mammary diseases characterised by long immunological and clinical latencies. To avoid these detrimental effects, Royal GD (GD), the Dutch Animal Health Organisation, started a voluntary SRLV accreditation in 1982 [4]. Throughout these years, GD has strived to continuously improve and fine-tune the conditions of this scheme in terms of diagnostics, practical implementation, traceability of animal movements, and information and communication technology (ICT). Accreditation is based on sequential herd screening using an antibody ELISA, followed by a confirmatory test in case of specific antibody detection. In the accreditation process, herd screening is performed by testing individual sera from a sample of animals from the herd, using the screening ELISA. Positive sera are re-tested using another confirmatory ELISA. This voluntary accreditation scheme aims for population freedom from infection, as described by Aalberts et al. [3]. In short, (dairy) goat herds and sheep flocks of unknown SRLV status can opt to enter this scheme after 12 months of restricted contact with accredited goats and sheep only, by randomly blood sampling a substantial number of individuals from at least six months of age, using a sample size allowing up to 0.5% actual infected individuals with a confidence interval (CI) of 95%. The private veterinary practitioner acts as a certified independent sampler and is responsible for correct selection. Animal contact structures are automatically monitored by a continuous exchange of legally required animal data between the mandatory identification and registration system of the government and GD’s identification and registration systems. Taking the specificity of the ELISA into account, single reactors are allowed to be retested in the screening ELISA once within a four- to six-week time frame. With a favorable result, accreditation will follow. Thereafter, surveillance will be carried out bi-annually in herds of up to 40 goats and in sheep flocks. Larger goat herds have to test yearly as of 2017, before which it was bi-annually. Testing is performed on individuals of at least 12 months of age using a sample size allowing up to 2% actual infected individuals with a CI of 95%. Currently, serum samples from accredited herds are tested in pools of five in a customized dilution [5]. In case of a reactive pool, the five individual sera are retested in the same screening ELISA. If one or more of these come up as positive, a confirmatory ELISA is used. If there is no explanation for the viral introduction, a resampling four to six weeks later is allowed, according to the same surveillance schedule. To exclude potential false positives in (previously) accredited herds, once again taking the specificity of the ELISAs into account, up to two individual recurrent seropositives can be sacrificed to be tested for a virus by the post-mortem sampling of the lung, udder and spinal cord using two PCR tests [6,7]. If any of these samples harbour a virus, a loss of accreditation will occur; if not, the benefit of the doubt applies and accreditation is continued [3]. In addition, bulk milk testing seems to be a useful tool for identifying SRLV infection in dairy herds, and potentially even for early detection. Previously, GD investigated the feasibility of bulk milk testing by titrating ELISA-positive pooled milk samples in negative milk, and by investigating bulk milk samples in relation to herd SRLV status. In this paper, (retrospective) bulk milk testing is discussed.

In recent years, the dairy goat sector has had to deal with a number of SRLV outbreaks in accredited herds. That is, dairy goat herds that had been CAEV-accredited for consecutive years suddenly faced unfavorable SRLV-seropositive results during the annual surveillance. In 2012, there were 666 CAEV-accredited herds, rising up to 810 in 2022; however, the number of dairy goat herds remained stable at around 400 throughout these years. Since 75% of these dairy goat herds participated in the program, the average annual percentage of herds with a loss of accreditation within this dairy goat herd population was 1.17% (0.33–2.67%).

The aim of this paper is to describe GD’s experience with the loss of CAEV herd accreditation in large dairy goat farms, with an emphasis on herd characteristics, diagnostic approach, and specific follow-up scenarios. Infection dynamics using the Röst Freed Model [8] are discussed, as well as the possibilities of including bulk milk and strategic pooled milk sample testing. GD is of the opinion that one should not only celebrate successes, but also be open and honest regarding set-backs. Therefore, we encourage others to share their experiences in aiming to improve programs collectively, start research into the underlying mechanisms, and deepen the knowledge of the causative pathogens.

## 2. Results

### 2.1. Dairy Goat Herds with Loss of CAEV Accreditation between 2012 and 2022

Table 1 gives an overview of the thirty-eight included dairy goat herds, or more specifically, the number of adult goats over twelve months of age, the year and month of loss of accreditation, the number of seropositive animals, and the timing of positive bulk milk testing, when available. The herd size was determined at the end of 2021 since it was unfortunately not possible to retrospectively retrieve reliable data on the herd size at the time of loss of accreditation for years earlier than 2018. Still, herd size was included to give an impression of the magnitude of some of these farms. Bulk milk sampling was only made available from 2018 onwards. Furthermore, if available, a potential route of disease introduction is mentioned. As can be seen, data are scarce since most farmers did not know how the infection was introduced. Also, the current CAEV status (unknown, infected, in observation or re-accredited) is stated and, if known, how re-accreditation was achieved.

The month of loss of accreditation depends on the date of registration within the program, and consequently, no timeline analysis had taken place.

On affected farms, a huge range exists in the number of test-positive goats, varying from a single reactor (herds 19, 22, 27, and 31) to over 100 individuals (herds 1, 11, 33, and 34). Those single reactors had been confirmed by both resampling four to six weeks later and by confirmatory testing, and in some cases by positive tissue PCR (herd 27).

Only thirteen affected herds had used bulk milk testing. In addition, six farmers (herds 17, 18, 23, 33, 34, and 37) had the possibility to test monthly samples in retrospect, and found out that bulk milk ELISA results were positive one (17), four (18, 34), seven (37), nine (33), or even eleven months (23) prior.

Although in the majority of affected herds route of introduction still remains unclear, purchase of goats originating from herds that in retrospect had a CAE infection was most likely in at least six cases (herds 7, 13, 21, 25, 28, and 38). In four cases (herds 3, 5, 15, and 26), contact with SRLV-unaccredited or even known-infected herds was a likely source of infection. In one case (herd 18), the farmer suspected that a substantial spill-over from the milk truck that had just come from another farm with a known CAE infection could have been the cause. The owner of herd 11 suspected the goat buyer who had entered his farm with a truckload of unaccredited goats, and had backed-up his trailer into the stable, as the possible source of infection. Finally, the owner of herd 14 had several dairy goat farms, and not all of them were accredited at the time, but staff and equipment were shared between locations.

The second-last column of Table 1 shows the current CAEV status with twenty-six herds (1, 3–13, 21–27, and 32–38) still infected, two with status unknown (herds 17 and 20), two in observation (herds 28 and 31) and eight re-accredited (herds 2, 14–16, 18–19, 29–30). The last column states how re-accreditation was achieved, with the purchase of a new CAE-accredited herd in two instances (herds 12 and 18), in one case being the second time (herd 12), and in two cases with the motherless rearing of their own goat kids (herd 2 and 14).

In very rare situations, qPCR was used within the accreditation system. In six cases (herds 3, 6, 11, 27, 30, and 33), tissue PCRs of lung, udder and spinal cord were conducted. Two herds (3 and 6) that had two goats that came up as PCR-negative in 2012 and 2014, respectively, were consequently considered CAEV-accredited but lost accreditation in 2015 and 2016. Another herd (11) had a similar situation with one PCR-negative in 2012, and a loss of accreditation in 2017. In 2020, herd 27 had one goat that was PCR-negative and one goat that was positive, and herd 30 tested one goat that was PCR-negative. Finally, herd 33 lost accreditation in 2020, and in 2022 had five PCR-positive goats.

### 2.2. (Retrospective) Bulk Milk Testing

In six cases, monthly bulk milk samples were tested by antibody ELISA (herds 1, 8, 18, 23, 26, and 37) as shown in Figure 1. From three of these herds (18, 23, and 37), stored milk samples from the year preceding the unfavourable test results were available and analysed to see how the tipping point was between the last favourable surveillance test results and the recent unfavourable surveillance test results. The change-over is somewhere in between these two surveillance moments, showing that infection and consequent seroconversion had taken place four (herd 18), nine (herd 37) and even eleven (herd 23) months earlier.

## 3. Discussion

The Dutch dairy goat industry has undergone enormous development in a few decades, with numbers increasing from an average of a few dozen animals per farm in the 1980s to an average of two hundred at the end of the 1990s, and finally to more than 1400 on average per farm in 2020 [1,9]. With this development, the demand for veterinary care also increased; when more became known regarding the harmful effects of CAEV infections over the course of the 1980s, GD started a CAEV accreditation program that made good use of the experiences of the MVV program that had started in 1982 [4]. Although GD has strived to continuously improve and fine-tune the conditions of this program [3], loss of accreditation could not always be prevented. With herds that lost accreditation, usually inexplicably and with sometimes large numbers of seropositive animals per farm, GD has explored possible ways to detect these seroconversions earlier. That is why it was decided to test herds of more than 40 goats annually from 1 January 2017 onwards, and why voluntary bulk milk testing was made available from 2018 onwards.

The average number of adult dairy goats over 1 year of age in the 38 herds that lost accreditation is 1686, which is larger than the average total number of 1411 goats on all 400 dairy goat farms in 2020 [1]. With a replacement rate of 25%, the expected average number of adult dairy goats on all farms is 1058, indicating that the mainly larger herds lost accreditation. In these herds, within-herd transmission can be impetuous as a consequence of frequent and intense contact, confirming the results of the Röst Freed Model depicted in Table 2. In housed sheep, the transmission rate of MVV, like CAEV belonging to the SRLVs, was about 1000 times faster than in sheep at grass, where transmission was negligible [10]. This was most probable, as CAE is also mainly a disease of housing wherein goats are kept in close proximity.

The number of positive ELISA results ranged significantly, from a single reactor to 140 out of 145 sampled goats. As shown in Table 2, after the introduction of a single CAEV infected goat, the rate of the spread of infection highly depends on the number of susceptible individuals and herd size. In small herds, it takes years for a CAEV infection to cause a serious impact. In the affected herds described in this paper, herd size ranged from 394 to 4589 adult dairy goats; thus, in theory, the whole herd could be infected within three years, based on the assumption that all goats have the same chance of contact with any other goat within the population. The herds with low numbers of reactors had possibly contracted CAEV relatively recently, and the herds with large numbers of reactors possibly earlier. The introduction of CAEV by a number of infectious goats could speed up infection dynamics. Furthermore, only a relatively small proportion of the herd is tested yearly, as surveillance is based on testing goats of at least 12 months of age using a sample size allowing up to 2% actual infected individuals with a CI of 95%. If the sampling strategy is carried out correctly and includes goats from all groups, reliable results could be expected—but this is not always the case, sometimes unintentionally, but also intentionally, for example when only sampling older goats with the best intentions. In at least two cases, only yearlings were seropositive, and the infections would have been missed if these had been excluded.

In herds with seropositive animals, retrospective bulk milk testing, which revealed seroconversion from four to eleven months prior to surveillance testing, confirmed its added value. Bulk milk represents all lactating goats of a herd and rules out incorrect sampling. Farmers can voluntarily subscribe to bulk milk testing, and logistics are all provided for samples being taken during milk collection.

Table 1 shows that it is difficult for large herds to become re-accredited: twenty-six out of the thirty-eight herds were still infected, and the two herds with status unknown and both herds with status in observation did not succeed in becoming re-accredited. Of the eight herds that regained accreditation, the two that did so by motherless rearing had several locations available. They were able to snatch kids at birth, feed them artificial colostrum and ship them off to another location with separate staff and equipment. In two cases, it is known that total-herd replacement had taken place by the purchase of CAEV-accredited replacements.

Successfully controlling SRLV depends largely on the optimal use of appropriate diagnostic tools. Immunological and clinical latencies hamper the diagnostic efficacy in SRLVs. The immunological latency causes a so-called serological gap [3], and the existence of an epidemiological latency (a time delay between infection and infectiousness) apparently exceeds the serological gap, as is demonstrated with a mathematical epidemiological model in sheep [10]. Once inside the host cell, innate immunity may interfere with SRLV replication, but the virus develops counteraction mechanisms to escape, multiply and survive [11]. Furthermore, antibody levels fluctuate throughout infection [11]. As a result, infection can remain under the radar. In spite of this, from a diagnostic point of view, SRLV ELISAs are the most appropriate tests in accreditation programs, but have limitations in their application for the confirmation of clinical cases [12].

Previously, GD investigated bulk milk samples in relation to herd SRLV status. Promising results were obtained with a putative detection limit of <3% within-herd prevalence using 1/10 pre-diluted samples, and even <1% within-herd prevalence when samples were tested undiluted. Moreover, out of the 249 bulk milk samples, all SRLV-accredited herds (n = 138) tested negative, while 50% of the samples from herds with an unknown SRLV status (n = 111) were positive [7]. At that time, bulk milk testing, although potentially useful, was not implemented in The Netherlands. More recently, retrospective testing of stored bulk milk samples after unfavourable diagnostic outcomes in yearly accreditation surveillance showed interesting results—specifically, clear serological tipping points presumably shortly after a rapidly spreading SRLV infection in a herd. As can be seen in Figure 1, the value of the bulk milk ELISA increases relatively quickly after the turning point. By frequently bulk tank milk testing in between the mandatory annual surveillance tests, seroconversions might be found closer to a possible turning point. In the earlier-mentioned GD study, the specificity of the bulk milk ELISA was 100%, with a cutoff of 0.25 [7]. Receiver operating characteristics analysis showed that with a lower cutoff of 0.10, specificity would still be >99% (data not shown). An analysis of GD data in 2020 showed that at least six herds that became bulk-milk-ELISA positive (cutoff 0.25) could be detected earlier, with a cutoff of 0.10. Therefore, GD decided to lower the cutoff to 0.10.

Although several attempts have been made to improve the control or eradication of SRLVs by performing PCR in blood samples, it is our experience that a PCR only has a limited added value, and merely in tissue samples but not in blood samples.

After loss of CAEV accreditation, different follow-up scenarios are discussed depending on the farmers’ wishes and possibilities. In determining potential future goals, finances play an important role. Since milk yield will most likely decrease as infection progresses, and production is the number one reason for replacements, the turnover of does will increase. Live (international) trade is also impacted, as goats from accredited herds represent a higher value. Goals might range from no further action and allowing CAEV to become endemic, to levels of control of spread, to total eradication. Once goals have been set, a tailor-made approach is proposed. To GD’s experience, the three main chosen goals are: (A) do nothing and start living with CAEV, (B) accept being non-CAEV-accredited and keep infection pressure low, and (C) aim at regaining CAEV-accreditation. In the latter case, several options are discussed: (1) total herd replacement, (2) artificial rearing of kids, (3) test and cull, and finally, (4) a combination of the three previous options. In general, high biosecurity levels, preferred ‘closed farming’ and in-case replacements should be bought in; quarantine and additional testing, even when goats are purchased from accredited farms, are important, as the consequences of the introduction of CAEV are enormous. Attention for other iceberg diseases such as CLA, paratuberculosis and border disease, and also zoonotic diseases such as chlamydiosis and salmonellosis underline this importance.

In large dairy goat herds kept inside, CAEV does not always seem to behave like a lentivirus, and transmission can be very fast as demonstrated in this paper. It is essential for CAEV-accredited herds to detect a possible virus introduction as soon as possible to increase the change of re-accreditation. Based on sensitivity and specificity, the bulk milk ELISA with a cutoff of 0.10, as described in this paper, is a valuable and cost-effective tool for the early detection of CAEV infection.

## 4. Materials and Methods

### 4.1. Study Population

Thirty-eight herds that lost CAEV accreditation between 2012 and 2022 were followed up. The characteristics of these herds were acquired from the legally required national identification and registration data base supervised by The Netherlands Enterprise Agency, and the customer relationship management system of GD. Size of affected herds ranges from 394 to 4589 adult dairy goats. Nine herds had less than 1000 dairy goats, eighteen between 1000 and 2000, four between 2000 and 3000, and four up to 4000, with one ranging over 4000.

Following unfavourable laboratory test results, these 38 farms were visited by a small ruminant GD veterinarian. Depending on the availability of samples and/or consent of the farmer, additional samples were tested.

### 4.2. Diagnostic Approach Antibody ELISAs and PCR

The Elitest MVV/CAEV ELISA (Hyphen Biomed, Neuville sur Oise, France) was used as a screening test, and the LSIVet Ruminant MAEDI-VISNA/CAEV Serum ELISA Kit (Thermo Fisher Scientific, Waltham, MA, USA) served as the confirmatory test. Based on specificity, sensitivity, and the possibility for sample pooling, these ELISAs were considered to be the most suitable ELISAs for accreditation purposes (according to fitness) for commercially available SRLV antibody tests in The Netherlands [3]. Sera from accredited herds were tested in pools of five in a customised dilution (1:100), applying a cutoff of 0.7. Sera from positive pools and from non-accredited herds were tested individually according to the manufacturer’s instructions, and positive individual sera were tested in the confirmatory ELISA according to the manufacturer’s instructions.

Bulk milk testing has been made available from 2018 onwards on a voluntary basis, since it is not included in the program regulations. Bulk milk testing can be performed as a precautionary measure, in addition to the program. Moreover, it can be applied as a follow-up tool after loss of accreditation. In the latter case, strategic pooled milk for groups housed together, specific age groups, exposed groups, groups at risk, or any chosen number of individual milk samples pooled can be tested.

(Bulk) milk samples were preserved with sodium azide. For retrospective testing, monthly taken bulk milk samples from the preceding year were stored at −20 °C until analysis. After unfavourable serum test results, bulk milk samples were analysed to identify the tipping point between the last favourable surveillance outcome and the unfavourable results. Bulk milk samples were tested with the Elitest MVV/CAEV ELISA according to the manufacturer’s instructions, using a cutoff of 0.1.

Udder, spinal cord and lung tissue samples of seropositive animals were tested using a qPCR, as described by Brinkhof et al. [6], until 2020. After 2020, tissue samples were tested using both genotype-A-specific and genotype-B-specific qPCR, as described by Michiels et al., 2018 [13].

### 4.3. Infection Dynamics of Caprine Arthritis Encephalitis Virus

A Röst Freed Model describes the dynamics of virus transmission in a herd, dependent on herd size [10]. GD applied this model on the dynamics of SRLV infection. For this model, a number of assumptions were made; for example, in theory, all individuals are susceptible, meaning they are capable of contracting the disease. After introduction of CAEV into a free population, a certain proportion can be designated as exposed (meaning infected but not yet infectious), and another proportion as infected and capable of transmitting the disease. When clinical signs of the disease become evident, it is assumed that these individuals can be characterised as infective, and will stay infective for the rest of their lives. Since CAEV causes a lifelong infection, meaning there is no clearance of infection, it is assumed that a population does not have any or only a negligibly small proportion of so-called recovered individuals that are permanently immune. It is assumed that all goats have the same chance of contact with any other goat within the population. Furthermore, it is assumed that the chance of effective contact resulting in transmission of infection is 2.5%. Putting these assumptions into the Röst Freed Model [8], infection can spread as depicted in Table 2.

### 4.4. CAEV Herd Status and Follow-Up Scenario’s

Four different herd statuses are possible within the CAEV program: unknown, infected, in observation, and accredited. Herds with a status unknown have never participated or no longer participate, and therefore no information regarding SRLV is available. After seropositives have been confirmed, a herd is considered SRLV infected. The status in observation is used for previously accredited herds that aim for re-accreditation after loss of accreditation. Accredited herds have gone through all stages of the accreditation process with a favourable result.

Follow-up scenarios start with a voluntary farm visit by a specialised small ruminant GD veterinarian, preferably together with the farm’s own veterinary practitioner. During this initial visit, CAEV backgrounds are shared; possible routes of infection and future impact on farm management as well as finances are discussed; and potential future goals are determined. The farmers are urged to inform owners of herds which have been in contact and therefore are at risk, to provide them with the opportunity to actively act on this information.

## 5. Conclusions

The characteristics of dairy goat herds with loss of CAEV accreditation show that larger herds seem to be at increased risk. Furthermore, re-accreditation seems more difficult for those larger herds, since only 8 out of the 38 herds acquired re-accreditation. Therefore, CAEV introduction should be avoided at all costs, for instance by biosecurity measures. Bulk milk testing demonstrates potential as a cost-effective tool for the early detection of CAEV infection in dairy herds. In addition to the yearly surveillance scheme, frequent bulk milk testing enables farmers to keep a close eye on potential SRLV introduction in their herd. By noticing a so-called tipping point, an alarm bell should ring that stimulates farmers to perform further investigation. Swift action could potentially reduce viral spread within and between herds, from which not only the farmer but also the industry as a whole could benefit. GD calls on others encountering similar problems to also publish these so that, collectively, programs can be improved.

## Figures and Tables

**Figure 1 pathogens-11-01541-f001:**
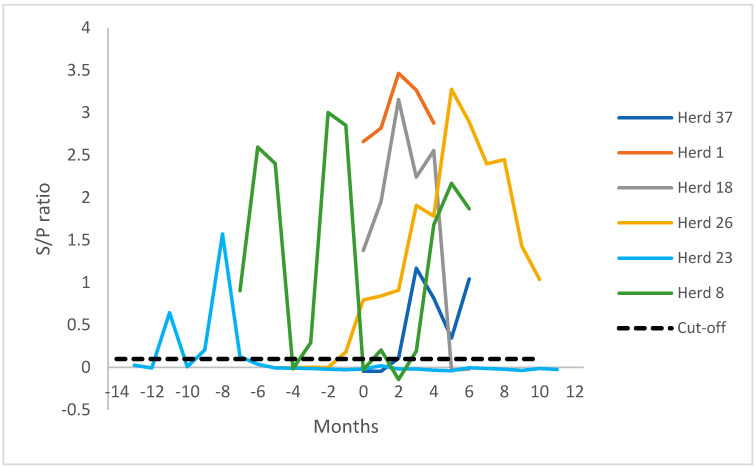
Bulk milk testing for caprine arthritis encephalitis virus antibodies (ELISA) in herds that lost CAEV accreditation.

**Table 1 pathogens-11-01541-t001:** Characteristics of dairy goat herds with loss of caprine arthritis encephalitis accreditation in the period 2012–2022 in The Netherlands.

Herd	No. of Goats > 1 Year (1 December 2021)	Date of Loss of Accreditation	No. of Positives/Negatives *	Bulk Milk Positive **	Potential Route of Introduction	Current Status	Return to Accreditation
1	1954	November 2013	116/24	May 2018	Unknown	Infected	n.a.
2	2905	January 2014	12/132	Not tested	Unknown	Accredited	Motherless rearing
3	2178	November 2015	2/140	Not tested	Sheep contact	Infected	n.a.
4	1215	December 2015	6/131	June 2018	Unknown	Infected	n.a.
5	716	January 2016	6/124	Not tested	Sheep contact	Infected	n.a.
6	645	May 2016	33/95	Not tested	Unknown	Infected	n.a.
7	790	July 2016	13/25	March 2019	Purchase	Infected	n.a.
8	1288	September 2016	11/127	September 2018	Unknown	Infected	n.a.
9	1277	September 2016	36/83	Not tested	Unknown	Infected	n.a.
10	1037	October 2016	19/116	Not tested	Unknown	Infected	n.a.
11	1253	January 2017	126/12	Not tested	Contact via trader	Infected	n.a.
12	1365	February 2017	Bulk milk	Not tested	Unknown	Infected	New herd (2007 and 2017)
13	1951	July 2017	6/135	June 2018	Purchase	Infected	n.a.
14	4589	July 2017	34/112	September 2018	Contact other herds	Accredited	Motherless rearing
15	645	July 2017	5/123	Not tested	Sheep contact	Accredited	n.a.
16	788	October 2017	2/134	Not tested	Unknown	Accredited	n.a.
17	1370	September 2018	33/105	August 2018	Unknown	Unknown	n.a.
18	1046	October 2018	40/101	July 2018	Spill milk truck	Accredited	New herd
19	1267	October 2018	1/136	Not tested	Unknown	Accredited	n.a.
20	541	November 2018	3/130	Not tested	Unknown	Unknown	n.a.
21	1029	January 2019	12/128	Not tested	Purchase	Infected	n.a.
22	984	June 2019	1/130	Not tested	Unknown	Infected	n.a.
23	3381	September 2019	3/141	August 2018	Unknown	Infected	n.a.
24	1695	September 2019	6/134	Not tested	Unknown	Infected	n.a.
25	1842	Novovember 2019	11/131	Not tested	Purchase	Infected	n.a.
26	2199	March 2020	36/107	March 2020	Sheep contact	Infected	n.a.
27	2374	April 2020	1/144	Not tested	Unknown	Infected	n.a.
28	3254	June 2020	6/150	Not tested	Purchase	In observation ***	n.a.
29	1790	July 2020	9/134	Not tested	Unknown	Accredited	n.a.
30	443	August 2020	7/114	Not tested	Unknown	Accredited	n.a.
31	878	October 2020	1/133	Not tested	Unknown	In observation ***	n.a.
32	394	September 2020	21/312	Not tested	Unknown	Infected	n.a.
33	3147	November 2020	140/5	March 2020	Unknown	Infected	n.a.
34	1463	March 2021	103/37	December 2020	Unknown	Infected	n.a.
35	1558	July 2021	5/142	Not tested	Unknown	Infected	n.a.
36	3789	August 2021	5/141	Not tested	Unknown	Infected	n.a.
37	3527	March 2022	22/125	September 2021	Unknown	Infected	n.a.
38	1530	June 2022	2/138	Not tested	Purchase	Infected	n.a.

* Positive and negatives samples add up to the total number of samples taken. ** Bulk milk testing available since 2018. *** “In observation” is status used for previously accredited herds that aim for re-accreditation after loss of accreditation. Not applicable (n.a.).

**Table 2 pathogens-11-01541-t002:** Numbers of infected animals in the years after introduction of caprine arthritis encephalitis virus infection in relation to different herd sizes.

Herd Size	Years after CAEV Introduction into a Herd				
	0	1	2	3	4
50	1	2	4	6	8
100	1	4	26	74	91
200	1	6	105	199	200
500	1	14	151	500	
1000	1	16	483	1000	
5000	1	126	4795	5000	

## Data Availability

The data in this study are presented as completely as possible but cannot be traced back to specific herds and are not publicly available due to privacy regulations. Background information is available on request from the corresponding author.

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
