# Peer review of "Loss of Caprine Arthritis Encephalitis Virus (CAEV) Herd Accreditation: Characteristics, Diagnostic Approach, and Specific Follow-Up Scenarios on Large Dairy Goat Farms"

_pathogens, 2022, doi:10.3390/pathogens11121541_

Round 1
Reviewer 1 Report
Dear author,
I found this manuscript interesting and well written. It is important for other goat countries to take part of the experiences from the Dutch CAE program. However, I think it would be better to publish it as a case report and not a scientific paper. For me it is not clear whether all your data is just a description of what is included in the program or if it is part of a separate study. For the latter, it needs a more detailed description to be published in a scientific paper. Moreover, there are no description of statistical analyses supporting some of your findings, for instance the use of bulk milk. The Röst Feed Model should also have been described in MoM. Finally, I am not convinced that all your conclusions have been explored in your study.
Author Response
Dear reviewer,
Thank you very much for your compliments and spending your valuable time in reading and reviewing our manuscript. This manuscript has been drawn up on invitation for a Special Issue entitled "Small Ruminant Lentiviruses (SRLVs): Genetic Diversity, Pathogenicity and Diagnostic Approach" of the journal Pathogens. This manuscript focusses on the diagnostic approach of CAEV. Pathogens does not have a case report section in this special issue. As authors we have discussed the issue if the work should better suit as an extended case report, but since 38 cases over a decade are involved we hope with implementing the other suggested changes the manuscript is now suitable as full paper. The data in our manuscript involves retrospective ordering of characteristics regarding loss of CAEV accreditation. Results are therefore not based on real statistics. As suggested the Rost Freed Model is placed in the MoM section were it belongs. The conclusions have been toned down to be confirm the results.
As organisation responsible for the accreditation scheme for CAE, we frequently get questions about (possible) reasons for loss of status for CAE. Since the dairy goat industry has developed very rapidly, with an average of approximately 1,400 dairy goats per farm, in the Netherlands in the last two decades, and the fact that CAEV is in many countries an important so called “Iceberg disease” in goats, we think that our experience on this topic is valuable for colleagues in other countries.
Reviewer 2 Report
The publication unfortunately has a lot of drawbacks because much of the data is inaccurate or missing and the paper is descriptive. The authors have shown nothing new. Many works have previously shown that herd size is an important risk factor for the spread of the MVV and CAEV.
Specific comments:
1. The abstract lacks results and conclusions.
2. The text starting from line 88 to 98 is a broader repetition of what is written in lines 99-103. The authors should modify this text.
3. Information on how many samples were tested in each herd is missing.
4. PCR results are missing. In the materials and methods authors declared that tested seropositive animals with qPCRs.
5. Lines 106-116. This is a description of what is shown in Table 1. The authors should avoid doing such things. Moreover, information on herd size and bulk milk sampling should be rather included in Material and Methods not in Results.
6. It is not explained what the abbreviation TM found in Table 1 means. Moreover, there should be no blank cells in the table 1. If there is no data or available information then this should be noted and placed in the table.
7. lines 150-163. These are not results and this text should be removed from the Results section. In addition, one more comment that both asymptomatic and symptomatic animals can transmit the virus.
8. " Furthermore, it is assumed that the 161 chance of effective contact resulting in transmission of infection is 2,5%."- citation is missing
9. What results are presented in Table 2. Are the results of the research described in the presented paper?
10. Figure 1 is not clear. It would be better to show only the results of flocks 18, 37 and 23 and it would be good to indicate when a flock lost accreditation.
Author Response
Dear reviewer, thank you very much for spending your valuable time in reading and reviewing our manuscript. This manuscript has been drawn up on invitation for a Special Issue entitled "Small Ruminant Lentiviruses (SRLVs): Genetic Diversity, Pathogenicity and Diagnostic Approach" of the journal Pathogens.
As authors we agree that this paper is mainly descriptive, and that risk factors have previously been described. Unfortunately it was not possible to follow-up all cases due to differences in availability of samples. Availability depended on whether or not samples we re stored, and farmer’s consent to test these or take additional samples. The Netherlands is a major player in the dairy goat industry, and many other (European) countries come to us for veterinary advice specifically of the implementation and experience on accreditation programs. Therefore we feel the urge to report on how we handle potential issues. For example herd specific advice after loss of accreditation. Also in our opinion (retrospective) testing of bulk milk sampling in relation to (loss of) herd accreditation is new.
Nevertheless, since the Dutch dairy goat industry has developed very rapidly, with an average of approximately 1,400 dairy goats per farm, in the last two decades, and the fact that CAEV is in many countries an important so called “Iceberg disease”, we think that our experiences on this topic are valuable for colleagues in other countries. We hope that with other proposed changes, you will find the manuscript suitable for publication. Below you find our response to your comments.
Specific comments:
- The abstract lacks results and conclusions.
Added: “Larger herds tended to be more prone to lose herd accreditation and had more difficulties regaining accreditation.” And “Special attention is paid to the added value of retrospective bulk milk testing or specific testing of groups based on the farm housing and management, which enables investigation of the moment and route of transmission to a herd.” And “intensifies surveillance and enables taking swift action to prevent further transmission within and between herds”
- The text starting from line 88 to 98 is a broader repetition of what is written in lines 99-103. The authors should modify this text.
We agree that these lines are more or less the same and therefore we removed lines 88-98.
- Information on how many samples were tested in each herd is missing.
Maybe we misunderstand your remark, since we think that in Table 1 the number of positive and negative samples per farm are mentioned. If you add these up you get the total number of sampled goats. As explanation, a footnote has been added to Table 1.
- PCR results are missing. In the materials and methods authors declared that tested seropositive animals with qPCRs.
Added: In very rare situations, qPCR was used within the accreditation system. In six cases (herds 3, 6, 11, 27, 30, and 33) tissue PCR’s of lung, udder and spinal cord have been conducted. Two herds (3 and 6) had two goat that came up as PCR negative in 2012 and 2014 respectively, were consequently considered CAEV accredited, but lost accreditation in 2015 and 2016. Another herd (11) had a similar situation with one PCR negative in 2012, and loss of accreditation in 2017. In 2020 herd 27 had one that was PCR negative and one positive, and herd 30 tested one that was PCR negative. And finally herd 33 lost accreditation in 2020 and in 2022 had five PCR positive goats.
- Lines 106-116. This is a description of what is shown in Table 1. The authors should avoid doing such things. Moreover, information on herd size and bulk milk sampling should be rather included in Material and Methods not in Results.
To the best of our knowledge depicted results should also be described in the text. Moreover, in lines 109 to 114 also context is given to describe the origin and background of the information. If our assumption is incorrect, these lines can easily be removed.
Herd sizes are described in the M&M section, but also mentioned in Table 1 to link this information to other herd characteristics with the aim to provide a full overview on available information in this Table. Moreover, this part of the M&M section has been made more concise and to the point.
- It is not explained what the abbreviation TM found in Table 1 means. Moreover, there should be no blank cells in the table 1. If there is no data or available information then this should be noted and placed in the table.
- lines 150-163. These are not results and this text should be removed from the Results section. In addition, one more comment that both asymptomatic and symptomatic animals can transmit the virus.
In agreement with the reviewers’ suggestion we replaced these lines from the results section to the introduction of the model to the M&M section. We agree that both asymptomatic and symptomatic animals can transmit the virus. In the discussion we have connected the model to our results.
- " Furthermore, it is assumed that the 161 chance of effective contact resulting in transmission of infection is 2,5%."- citation is missing
This is our estimate to the best of our knowledge.
- What results are presented in Table 2. Are the results of the research described in the presented paper?
Indeed, you are right, this is not a result per se. This is the model that GD has used where we have made the assumptions as now described in the M&M section. Amendments have been made. See our reply to comment 7.
- Figure 1 is not clear. It would be better to show only the results of flocks 18, 37 and 23 and it would be good to indicate when a flock lost accreditation.
Thanks you for this suggestion. We have changed the layout of figure 1. Results of retrospective bulk milk samples have been timed “in the past” (before timepoint 0). Thereby the retrospective result can be (mostly) separated from the follow-up results (herds 1, 8, and 26).
Reviewer 3 Report
Dear authors,
thank you very much for this interesting manuscript.
I would like to submit to your attention some suggestions, as following.
PAGE 2, LINES 56-80: generally, I found the description of accreditation and re-accreditation processes quite confusing, even after I read the reported references (n. 5 particularly). For instance:
· lines 56-57: in the accreditation process, authors said the first screening ELISA is followed by a confirmatory test but later it seems that the confirmatory test is used for surveillance test after farm accreditation (lines 74-76);
· lines 67-69: when a single reactor is tested again, after 4 or 6 weeks from the first time, what kind of ELISA is used? I presume ELISA screening, but it seems not immediately clear;
· lines 74-76: in the surveillance test after accreditation, serum samples are tested in pools and, in case of a reactive pool, the five individual sera are tested again. Authors said confirmation ELISA test is used, but I presume that ELISA screening is also used firstly, to identify exactly which sample/samples is/are positive between the five samples in the pool. Instead, in Material and Methods paragraph (lines 312-314) it is clearer.
· lines 77-79: PCR test on post-mortem samples. Reactive animals in serological tests are culled because positive, and then PCR tests are carried out; or these animals are kept in the farms until they die naturally? I understand this last one, but in this case, at least a seropositive animal is kept in the farm for others months, delaying the end of the accreditation process.
· lines 79-80: what do you mean with “the benefit of the doubt applies”? these farms are considered free or not?
PAGE 2, LINES 85-89: between 2012-2022, 38 accredited farms showed unfavourable serological results during surveillance. in the same years, how many farms entered in the accreditation voluntary programme and obtained accreditation? in other words, what is the percentage of farms that lost accreditation out of the total number of companies that participated in the program, in years 2012-2022?
PAGE 3, LINE 102: it is mentioned “strategic pooled milk sample”, but in the paper only bulk milk is described as a useful tool to identify a tipping point.
PAGE 5, LINES 172-180; PAGE 8, LINES 315-318: bulk milk samples are taken monthly during the accreditation process in all farms? or only in some farms? why?
PAGE 9, LINES 328-329: if a herd is considered SRLV infected after seropositive animals are confirmed, what is the value of the PCR test in this system?
Minor edits required:
LINE 46: please, add (CAE) after “caprine arthritis encephalitis”;
LINES 47-48: please, add (SRLV) after “The small ruminant lentiviruses”;
LINE 52: please, explain what is GD;
LINE 126: a single reactor in also in farm 27, according to table 1;
LINES 129-132: these data are not in table 1, right?
LINES 135-136: contact with sheep for farms 3, 5, 15 and 26 (not 25);
LINE 145-146: in observation farm 31 (not 32), according to table 1;
LINE 146: re-accredited farm from 29-30 (not 20), according to table 1;
TABLE 1: in correspondence of farm 12, what is “TM”?
LINE 172: please, check where bracket is placed: (.Retrospectiv)e
FIGURE 1: I found it not so clear: I don’t understand where the tipping points are, relating to months.
Author Response
Dear reviewer, thank you very much for spending your valuable time in reading and reviewing our manuscript. This manuscript has been drawn up on invitation for a Special Issue entitled "Small Ruminant Lentiviruses (SRLVs): Genetic Diversity, Pathogenicity and Diagnostic Approach" of the journal Pathogens.
PAGE 2, LINES 56-80: generally, I found the description of accreditation and re-accreditation processes quite confusing, even after I read the reported references (n. 5 particularly). For instance:
Lines 56-57: in the accreditation process, authors said the first screening ELISA is followed by a confirmatory test but later it seems that the confirmatory test is used for surveillance test after farm accreditation (lines 74-76);
We changed the text according your suggestion into: In the accreditation process, herd screening is performed by testing individual sera from a sample of animals from the herd, using the screening ELISA. Positive sera are re-tested using another, confirmatory ELISA. After accreditation, surveillance is performed by testing pooled individual sera (5 sera per pool) from a sample of animals from the herd. Samples from positive pools are re-tested individually using the screening ELISA. Individual samples that are positive in the screening ELISA are re-tested using the confirmatory ELISA.
Lines 67-69: when a single reactor is tested again, after 4 or 6 weeks from the first time, what kind of ELISA is used? I presume ELISA screening, but it seems not immediately clear;
Amendment to the text: Taking the specificity of the ELISA into account, single reactors are allowed to be retested in the screening ELISA once within a four to six weeks’ time frame. In case of a reactive pool, the five individual sera are retested in the same screening ELISA. If these come up as positive, a confirmatory ELISA is used. If there is no explanation for viral introduction resampling four to six weeks later is allowed according to the same surveillance schedule.
Lines 74-76: in the surveillance test after accreditation, serum samples are tested in pools and, in case of a reactive pool, the five individual sera are tested again. Authors said confirmation ELISA test is used, but I presume that ELISA screening is also used firstly, to identify exactly which sample/samples is/are positive between the five samples in the pool. Instead, in Material and Methods paragraph (lines 312-314) it is clearer.
Lines have been amended to: “In case of a reactive pool, the five individual sera are retested in the same screening ELISA. If these come up as positive, a confirmatory ELISA is used. If there is no explanation for viral introduction resampling four to six weeks later is allowed according to the same surveillance schedule.”
Lines 77-79: PCR test on post-mortem samples. Reactive animals in serological tests are culled because positive, and then PCR tests are carried out; or these animals are kept in the farms until they die naturally? I understand this last one, but in this case, at least a seropositive animal is kept in the farm for others months, delaying the end of the accreditation process.
Amended to: “To exclude potential false positives in (previously) accredited herds, once again taking the specificity of the ELISAs into account, up to two individual recurrent seropositives can be sacrificed to be tested for virus by post-mortem sampling of lung, udder and spinal cord using two PCR tests [8,9].”
Lines 79-80: what do you mean with “the benefit of the doubt applies”? these farms are considered free or not?
Added: and accreditation is continued
PAGE 2, LINES 85-89: between 2012-2022, 38 accredited farms showed unfavourable serological results during surveillance. in the same years, how many farms entered in the accreditation voluntary programme and obtained accreditation? in other words, what is the percentage of farms that lost accreditation out of the total number of companies that participated in the program, in years 2012-2022?
Lines 88-98 are taken out of the introduction as suggested by another reviewer. To answer your question we have added: “In 2012 there were 666 CAEV accredited herds rising up the 810 in 2022, although the number of dairy goat herds remained stable at around 400 throughout these years. Since 75% of these dairy goat herds participate in the program, the average percentage of herds with loss of accreditation within this dairy goat herd population was 1,17% (0.33-2.67%)”.
PAGE 3, LINE 102: it is mentioned “strategic pooled milk sample”, but in the paper only bulk milk is described as a useful tool to identify a tipping point.
The following text has been included I the MoM section (page 8): “Bulk milk testing has been made available from 2018 onwards on a voluntary base since it is not included in the program regulations. Bulk milk testing can be performed as a precautionary measurement, in addition to the program. Moreover, it can be applied as a follow-up tool after loss of accreditation. In the latter case strategic pooled milk for example groups housed together, specific age groups, exposed groups, groups at risk, or any chosen number of individual milk samples pooled can be tested.”
PAGE 5, LINES 172-180; PAGE 8, LINES 315-318: bulk milk samples are taken monthly during the accreditation process in all farms? or only in some farms? why?
The following text has been included I the MoM section (page 8): “Bulk milk testing has been made available from 2018 onwards on a voluntary base since it is not included in the program regulations. Bulk milk testing can be performed as a precautionary measurement, in addition to the program. But it can also be applied as a follow-up tool after loss of accreditation. In the latter case strategic pooled milk for example groups housed together, specific age groups, exposed groups, groups at risk, or any chosen number of individual milk samples pooled can be tested.”
PAGE 9, LINES 328-329: if a herd is considered SRLV infected after seropositive animals are confirmed, what is the value of the PCR test in this system?
The following text has been included in line 77: “To exclude potential false positives in (previously) accredited herds, once again taking the specificity of the ELISAs into account, up to two individual recurrent seropositives can be sacrificed to be tested for virus by post-mortem sampling of lung, udder and spinal cord using two PCR tests [8,9].
Minor edits required:
LINE 46: please, add (CAE) after “caprine arthritis encephalitis”; done
LINES 47-48: please, add (SRLV) after “The small ruminant lentiviruses”; done
LINE 52: please, explain what is GD; included; Royal GD (GD), the Dutch Animal Health Service
LINE 126: a single reactor in also in farm 27, according to table 1; herd 27 has been added
LINES 129-132: these data are not in table 1, right? You are right. This information is only been mentioned in the text.
LINES 135-136: contact with sheep for farms 3, 5, 15 and 26 (not 25); corrected
LINE 145-146: in observation farm 31 (not 32), according to table 1; corrected
LINE 146: re-accredited farm from 29-30 (not 20), according to table 1; corrected
TABLE 1: in correspondence of farm 12, what is “TM”? TM has been replaced with Bulk milk
LINE 172: please, check where bracket is placed: (.Retrospectiv)e corrected
FIGURE 1: I found it not so clear: I don’t understand where the tipping points are, relating to months. We have changed the layout of figure 1. Results of retrospective bulk milk samples have been timed “in the past” (before timepoint 0). This also improves visibility of the tipping points (point of (sero)conversion).
Round 2
Reviewer 1 Report
Dear author,
My main objection with this manuscript remains. It is an excellent case report. Well written and clearly presented. For us working with lentiviruses, it is of great interest to get reports from different national programs. However, it is still not suitable as a scientific paper and I encourage you to submit it elsewhere as case report.
Author Response
Dear reviewer,
We are once again thankful for your detailed review of the manuscript. Your comments and advices have been of great aid and have contributed to substantially increase the quality of the manuscript. We have addressed the comments as best as we could, and seemingly satisfactory.
We agree that this is not about in-depth scientific research, but retrospective descriptive. But is about much more than a case report. As authors were are of the opinion that we should not only celebrate our successes, but also be open and honest on our set-backs. Therefore we see it as our duty to analyse those setbacks, and to describe those analyses in the same or similar scientific journals in which programs and tests are described. This should encourage others to do the same and not only improve these programs but also deepen knowledge of the causative pathogens. This may then leads to hypotheses for further research.
To reduce the possible confusion caused by us, we have adapted suggestions about a case report in the text.
We sincerely hope we can appeal to you to reconsider you position.
Best regards,
The authors
Reviewer 2 Report
The authors corrected what they could correct, however, the work still has its flaws (incomplete data, selectively done research), which cannot be corrected.
Author Response
Dear Reviewer,
We are once again thankful for your detailed review of the manuscript. Your comments and advices have been of great aid, although unfortunately issues such as of incomplete data remain.
It is true that, as you rightfully addresses, there are points for improvement. We have addressed the comments as best as we could, trimmed, reshuffled, and reconsidered the manuscript. We agree that this is not about in-depth scientific research, but retrospective descriptive about the best possible analysis of setbacks in a CAEV program that is based on scientific studies in the past decades. Unfortunately, when analysing setbacks you cannot build a study as you would like and we have to make do with the available data. Unfortunately we can't make it better.
Best regards,
The authors
Round 3
Reviewer 2 Report
I accept the authors' explanation.
Author Response
Dear Reviewer,
Thank you for your keen eye on this manuscript.
You are absolutely right in your comments. In the first case we have segregated the suggested changes in a later stage. (Both in the accreditation and in the surveillance process the same confirmatory test is used.)
The result is as follows:
Line 58-59: Accreditation is based on sequential herd screening using an antibody ELISA, followed by a confirmatory test in case of specific antibody detection.
Added in: In the accreditation process, herd screening is performed by testing individual sera from a sample of animals from the herd, using the screening ELISA. Positive sera are re-tested using another, confirmatory ELISA.
Part two of our suggested amendment (After accreditation, surveillance is performed by testing pooled individual sera (5 sera per pool) from a sample of animals from the herd. Samples from positive pools are re-tested individually using the screening ELISA. Individual samples that are positive in the screening ELISA are re-tested using the confirmatory ELISA) we think is best described as currently in the lines 78-81: Currently, serum samples from accredited herds are tested in pools of five in a customized dilution [5]. In case of a reactive pool, the five individual sera are retested in the same screening ELISA. If one or more of these come up as positive, a confirmatory ELISA is used.
2 more differences that I observed-I believe of no great importance-between manuscript and authors reply are the following:
- In the amendment to the lines 67-69 : instead of “these” the manuscript says “if one or more of these”.
True. This amendment has been suggested by one of my co-authors who pointed out that instead of only in case of up to five ELISA positive individual samples a confirmatory ELISA is used, this is nowadays the case for all of the positives. This is the result of an ICT related system change that unfortunately no longer could adhere to the restriction to five.
- Lines 88-98 : instead of “percentage”, the manuscript says “annual percentage”
Correct. One of my co-authors remarked that I calculated the (mean) annual percentage and should refer to these correctly.
Kind regards on behalf of the authors,
Karianne Peterson